



# A Universal Multifractal Approach to Assessment of Spatiotemporal Extreme Precipitation over the Loess Plateau of China

Jianjun Zhang[1, 3, 5], Hoshin V. Gupta[2], Guangyao Gao[3], Bojie Fu[3], Xiaoping Zhang[4] and Rui Li[4]

[1]School of Land Science and Technology, China University of Geosciences, Beijing 100083, China
5 [2]Department of Hydrology and Atmospheric Sciences, The University of Arizona, Tucson, AZ 85721, USA
[3]State Key Laboratory of Urban and Regional Ecology, Research Center for Eco-Environmental Sciences, Chinese Academy of Sciences, Beijing 100085, China
[4]State Key Laboratory of Soil Erosion and Dryland Farming on the Loess Plateau, Institute of Soil and Water Conservation, Chinese Academy of Sciences and Ministry of Water Resources, Yang ling, Shaanxi, 712100, China
10 [5]Key Laboratory of Land Consolidation and Rehabilitation, Ministry of Land and Resources, Beijing 100035, China

*Correspondence to*: Hoshin V. Gupta (hoshin@email.arizona.edu)

**Abstract.** Extreme precipitation (EP) is a major external agent driving various natural hazards in the Loess Plateau (LP), China. Yet, the characteristics of spatiotemporal EP responsible for such hazardous situations remain poorly understood. We integrate universal multifractals with a segmentation algorithm to characterize a physically meaningful threshold for EP (EPT). Using 15 daily data from 1961 to 2015, we investigate the spatiotemporal variation of EP over the LP. Our results indicate that, with precipitation increasing, EPTs range from 17.3 to 50.3 mm/d while the mean annual EP increases from 35 to 138 mm from northwest to southeast LP. Further, EP frequency (EPF) has historically spatially varied from 54–116 days, with the highest EPF occurring in the mid-southern and southeastern LP where precipitation is much more abundant. However, EP intensities tend to be strongest in the central LP where precipitation also tends to be scare, and get progressively weaker as we move 20 towards the margins (similarly with EP severity). An examination of atmosphere circulation patterns indicates that the central LP is the boundary where tropical cyclones reach furthest into inland China, resulting in the highest EP intensities and EP severities being in this area. Under the control of the East Asian monsoon, precipitation from June to September accounts for 72% of the total amount, while 91% of the total EP events are concentrated in June to August. Further, EP events occur, on average, 11 days earlier than the wettest part of the season. These phenomena are responsible for the most serious natural 25 hazards in the LP, especially in the Central region. Spatiotemporally, 91.4% of the LP has experienced a downward trend of precipitation, while 62.1% of the area has experienced upward trends of the EP indices, indicating the potential risk of more serious hazardous situations.

## 1. Introduction

Extreme precipitation (EP) is the dominant external agent driving floods, erosion, and debris flow etc., with adverse impacts 30 on human life, the social economy, the natural environment, and ecosystems (Min et al., 2011; Pecl et al., 2017; Walther et al., 2002). These impacts are especially severe in arid and semiarid areas because of the sparsity of vegetation and fragility of the





eco-environment (Bao et al., 2017; Huang et al., 2016). In recent decades, worldwide climate change has given rise to spatially heterogeneous changes in EP regime (Donat et al., 2016; Manola et al., 2018; Zheng et al., 2015). Such uneven changes in EP have the potential to aggravate adverse impacts on human life and eco-environment and, consequently, EP has recently

received increased attention (Eekhout et al., 2018; Li et al., 2017; Wang et al., 2017). In this regard, a fundamental need is to evaluate the regional spatiotemporal variation of EP, providing important information that is crucial to natural resource management and sustainable social development.

The Loess Plateau (LP), which is located in the middle reaches of the Yellow River, is a typical arid and semiarid region, characterized by serious EP-induced natural hazards, including soil erosion and consequent hyper-concentrated flooding and

occasional landslides, etc. (Cai, 2001). In the LP area, EP-induced soil erosion generates some of the highest sediment yields observed on earth, ranging from $3 \times 10^3$ to $4 \times 10^4$ t·km$^{-2}$·yr$^{-1}$. For example, sediment delivered to the Yellow River in recent decades was estimated to be $16 \times 10^8$ t·yr$^{-1}$ (Ran et al., 2000; Tang, 2004), and sediment deposition resulted in the river bed of the lower Yellow River aggrading by 8–10 m above the surrounding floodplain (Shi and Shao, 2000). As it flows on the aggraded thalweg, the extreme-precipitation-driven hyper-concentrated floodwaters cause, in turn, the lower Yellow River to

burst its channel. Over the past 2500 years, this has cause flooding 1,593 times, and changes to the course of the river channel 26 times, leading to unimaginable death and devastation (Ren, 2006; Tang, 2004). To control such EP induced natural hazards, ecological restoration projects have been implemented over the LP. For example, the "Grain for Green" project (the largest investment project in China) was implemented to control natural hazards such as soil erosion and flooding, and has cost more than 75 billion dollars over the past 20 years.

Accordingly, a better understanding of spatiotemporal EP changes in this area is of considerable interest for various fields, such as risk estimation, land management, flood control, and infrastructure planning (Feng et al., 2016; Wang et al., 2015). Considerable past work has been devoted to investigating the spatiotemporal variation of total precipitation and precipitation extremes in this region, with consensus obtained for precipitation "amount" (Li et al., 2010a; Li et al., 2010b; Miao et al., 2016; Wan et al., 2014; Xin et al., 2009). However, the spatial distribution of EP in the LP is still poorly understood, with considerable

disagreement regarding EP and the inability to account for the spatial distribution of EP-induced hazards such as soil erosion (Li et al., 2010a; Li et al., 2010b; Miao et al., 2016; Wan et al., 2014; Xin et al., 2009). It is therefore important to account for the spatiotemporal role of EP in natural hazards, to facilitate better catchment management in regards to issues such as freshwater shortage (Feng et al., 2012).

To understand spatiotemporal variations in EP, scientists are often required to collect more detailed data, including

maximum depth, duration, and area observations (Dulière et al., 2011; Herold et al., 2017; Miao et al., 2016). Despite sophisticated methodologies, such efforts rely on data from various sources, which are typically absent in the long-term historical observational records, especially over large areas. Therefore, any investigation of spatiotemporal variation in EP must make use of the information in available historical data that was observed at fixed time intervals (e.g. daily). However,





EP events tend to be relatively rare, unpredictable, and often of short duration (Liu et al., 2013), and this uncertainty, combined
with varying geographical and meteorological conditions, increases the complexity of EP assessment.

In general, existing methods for EPT determination can be grouped into two categories: nonparametric and parametric. Non-parametric methods use fixed critical values or percentiles to define the thresholds for extreme events. Because the corresponding classification of EPTs varies from region to region (e.g., a 50 mm daily precipitation event is considered normal in South China but would be an EP event in the LP), the application of non-parametric methods can require considerable
subjectivity (Liu et al., 2013), and significantly affect the results of the analysis. For instance, using an absolute value of 50 mm/d, Xin et al. (2009) reported a spatiotemporal decreasing zone of EP in mid-eastern LP while, using the 95% percentile to determine EP, Li et al. (2010a) and Li et al. (2010b) found an increasing trend of EP frequency in the southeastern LP. These reports did not, however, explain the rational for spatial variation of EP and its impacts on the most serious soil erosion in the central LP.

Parametric statistical methods based on empirical distributions have recently become popular. A variety of special distribution functions and parameter estimation techniques have been proposed to characterize observed EP (Anagnostopoulou and Tolika, 2012; Beguer á et al., 2009; Deidda and Puliga, 2006; Dong et al., 2011; Li et al., 2005; Pfahl et al., 2017). A recent focus that has emerged is to obtain better physical understanding of EPTs, and thereby to assess regional variations in EP. For example, Liu et al. (2013) adopted a multifractal detrended fluctuation analysis (MFDFA) to determine EPTs, and Du
et al. (2013) applied MFDFA to investigate EPTs and consequent EP variation in Northeastern China. To date, however, no international standards for the selection of such methods exist.

Recent investigations of precipitation using the universal multifractal technique have demonstrated its multifractal nature. Universal multifractals were conceived to study the multiplying cascades governing dynamics of various geophysical fields (Lovejoy and Schertzer, 2013; Schertzer and Lovejoy, 1987). For precipitation, a scaling break separates the meteorological
and climatological regimes at roughly two weeks (Tessier et al., 1996; Tessier et al., 1994, 1993). The meteorological scaling interval indicates that, from the multifractal perspective, data collected at time intervals of one day and those at intervals finer than minutes can equivalently characterize the physical processes associated with precipitation (Pandey et al., 1998; Tessier et al., 1996), indicating that EP events can, in principle, be characterized by the study of daily data observed at gauging stations. Of course, it is vitally important that the universal multifractal characterizes how extremes occur in a natural manner (Lovejoy
and Schertzer, 2007; Tessier et al., 1996).

In this study, we use the universal multifractal technique to obtain a physically meaningful characterization of EPT. Our objectives are to (1) apply the universal multifractal approach to determine a unique set of EPTs for the LP area, (2) investigate how spatial variations in EP are responsible for the severe nature of soil erosion, and (3) assess the spatiotemporal variation of EP over the LP during the period 1961-2015.



## 2 Methodology

### 2.1 The relationship between precipitation extremes and Multifractals

The approach outlined below was used to identify EP events at the observation time scale. In the method of universal multifractals, two equivalent routes can be followed to investigate time series scaling: the probability distribution and the statistical moments. A fundamental property of multifractal fields related to the probability distribution is given by the equation (Lovejoy and Schertzer, 2013; Schertzer and Lovejoy, 1987):

$$\Pr(\varphi_\lambda > \lambda^\gamma) \approx \lambda^{-c(\gamma)} \tag{1}$$

where $\lambda$ represents the resolution of the measure (i.e., the ratio of the external scale $L$ to the measurement scale $l$; $\lambda = L/l$), $\varphi_\lambda$ is the intensity of the field measured at resolution $\lambda$ (in this case the density of stations), $\gamma$ is the order of singularity corresponding to $\varphi_\lambda$, and the codimension function $c(\gamma)$ characterizes the sparseness of the $\gamma$-order singularities (this function is a basic multifractal probability relation for cascades). Accordingly, the statistical moments are given by

$$\langle \varphi_\lambda^q \rangle = \lambda^{K(q)} \quad \lambda > 1 \tag{2}$$

where $K(q)$ is the multiple scaling exponent function for moments. The two equivalent routes are related via a Legendre transform (Parisi and Frisch, 1985). The universal $K(q)$ functions and the codimension function $c(\gamma)$ are expressed as

$$K(q) = \begin{cases} \dfrac{C_1}{\alpha - 1}\left(q^\alpha - q\right) & \alpha \neq 1 \\ C_1 q \log(q) & \alpha = 1 \end{cases} \tag{3}$$

$$c(\gamma) = \begin{cases} C_1\left(\dfrac{\gamma}{C_1 \alpha'} + \dfrac{1}{\alpha}\right)^{\alpha'} & \alpha \neq 1 \\ C_1 \exp\left(\dfrac{\gamma}{C1} - 1\right) & \alpha = 1 \end{cases} \tag{4}$$

where $0 \leq \alpha \leq 2$ is the multifractal index, which describes how rapidly the fractal dimensions vary as we leave the mean. The term $C_1$, the codimension of the mean of the process, varies on $0 \leq C_1 \leq D$ ($D$ is space dimension; $D = 1$ for time series) and quantifies the sparseness of the mean. In this paper, the parameters $\alpha$ and $C_1$ of the multifractal model were estimated using the double trace moment technique (Lavallée et al., 1993).

As noted by Gagnon et al. (2006) and Lovejoy and Schertzer (2007), the parameters $C_1$ and $\alpha$ characterize the mean of the field, whereas the extremes are expressed by the singularity, $\gamma$, and the codimension function $c(\gamma)$ (Hubert et al., 1993). For $0 \leq \alpha < 1$ and considering a time series of infinite length, a finite maximum order of singularity, $\gamma_0$, can be determined as





$$\gamma_0 = \frac{C_1}{1-\alpha} \tag{5}$$

However, in general, any time series of finite length will almost surely miss the presence of rare extremes in the field. In
this case, the observed singularities will be bounded by an effective maximum singularity, $\gamma_s$:

$$\gamma_S = \gamma_0 \left[ 1 - \alpha \left( \frac{C_1}{D} \right)^{-1/\alpha'} \right] < \gamma_0 \tag{6}$$

The parameter $\gamma_S$ links the physical processes that generate precipitation events to the conceptual model of multiplicative cascades, and allows the extremes to be cast in a probabilistic framework.

**2.2 Determination of the extreme precipitation threshold**

The approach outlined below was used to estimate the EPTs and EP events for each station, using the singularity parameter, $\gamma_s$. Given that the multifractal parameters $\alpha$ and $\gamma_s$ naturally characterize extremes, both of these parameters will change if we gradually remove extreme values from the data set. The singularity of the precipitation data series will be completely changed, and the two parameters will significantly change if all of the extreme values are deleted. To obtain a physically meaningful value for the EPT, we attempted to estimate the multifractal parameter series by applying the universal multifractal approach
to our precipitation series and successively eliminating maximum values of precipitation. However, as shown by Figures 1a and 1c, the degree of convergence to the original value is not unique, with the values fluctuating slightly around the original $\alpha$ and $\gamma_s$. Accordingly, the variance series of index series $\alpha_j$ and $\gamma_{s,j}$ were computed to eliminate the fluctuation while identifying the point of convergence. The procedure is as follows:

1) Eliminate the data point $x_j, \{x_j, x_j \geq x_{max} - d \times j\}$ from the precipitation time series $\{x_i, i = 1, 2,..., n\}$, where $x_{max}$ is the
maximum, $x_{ave}$ is the average, and $d$ is the interval space.

2) Compute the selected parameters.

3) Repeat (1) and (2) for $j$ varying from 1 to int($(x_{max}-x_{ave})/d$).

Using the obtained parameter series, we applied the segmentation algorithm proposed by Bernaola Galván et al. (2001) to determine the point of abrupt change, which we define as the EPT. The segmentation algorithm is based on the calculation of
the statistic $t$ of each data point in a series or subseries:

$$t = \left| \left( \mu_L - \mu_R \right) / s_D \right| \tag{7}$$

The significance level $P(t)$ of the largest value $t_{max}$ obtained from Eq. (7) is defined as the probability of obtaining a value equal to or less than $\tau$ within a random sequence (Swendsen and Wang, 1987)





$$P(t) = \Pr\{t_{\max} \leq \tau\} \tag{8}$$

If the significance exceeds a selected threshold $P_0$ (usually taken to be 95%), an abrupt point is selected and the series is cut at this point into two subsets.

The pooled variances and the abrupt points of the $\alpha$ and $\gamma_s$ series are shown in Figures 1b-1d. The abrupt point, where $s_D$ differs from its left-side points but is approximately equal to its right-side points, is selected to be the EPT. As shown in Figure 1, the EPT for Xingxian station is estimated as 37.1 mm/d, and 90 EP events have occurred over the past 55 years (Figure 1c).

**2.3 EP indices**

All of the variables characterizing spatiotemporal EP over the LP are shown in Table 1. The crucial variable, EPT and the mean annual EP (MEP) are presented to provide a general description of EP. As noted by *IPCC (2007)*, the severity of EP (EPS) events relies on both intensity and frequency. Neither the cases of high frequency with low intensity, or low frequency with high intensity, can reflect the severity of EP events given a long-term time series for an area. Consequently, we examine

the extreme precipitation intensity (EPI), extreme precipitation frequency (EPF, defined below) and EPS to characterize the spatiotemporal nature of EP over the LP. In addition, we compute the mean daily precipitation (MDP) and the accumulated daily EP events (ADEP) to help characterize the intra-annual precipitation and EP.

The EPF at a station is the number of days exceeding EPT. Annual values of EPI ($P_I$) for a station are given by

$$P_I = \frac{1}{n}\sum_{i=1}^{n}\left(PE_i - P_T\right)\Big/P_T, \; i = 1, 2, ..., n. \tag{9}$$

where $P_T$ and $PE$ represent EPT and the magnitude of EP, respectively. To obtain the concordant EPS, each annual EPF/EPI series is standardized to the range 0 to 1 using equation (10):

$$X = (X_i - X_{\min})/(X_{\max} - X_{\min}) \tag{10}$$

where $X_{\min}$ and $X_{\max}$ represent the lowest and highest annual EP frequency/intensity, respectively. The annual EPS for each station ($0 \leq \mathrm{EPS} \leq 1$) is calculated from the standardized $P_I$ and $P_F$ by

$$EPS = k_1 P_I + k_2 P_F \tag{11}$$

where k1 and k2 are the weight coefficients of frequency and intensity influencing EP severity, respectively, and k1 + k2 = 1. In this paper, $k1$ and $k_2$ are set 0.5, as both EPI and EPF play major roles in the EPS of EP events [IPCC, 2007].


### 2.4 Spatiotemporal EP presentation

The spatial distributions of the EP indices were derived by interpolation via Kriging (Oliver and Webster, 1990), using data observed at the gauging stations. All spatial analysis were carried out using the ArcGis 10.1 software. Spatiotemporal trends for annual EP variables were computed for each pixel using the least squares method. Following Stow et al. (2003), the trend is defined as the slope of the least squares line that fits the inter-annual variability of individual EP indices during the study period, given by:

$$S = \frac{n \times \sum_{i=1}^{n}(i \times P_i) - \left(\sum_{i=1}^{n} i\right)\left(\sum_{i=1}^{n} P_i\right)}{n \times \sum_{i=1}^{n} i^2 - \left(\sum_{i=1}^{n} i\right)^2} \tag{12}$$

where $n$ is the total of years, $P_i$ is value of the pixel in the $i$-th year.

### 3. Study area and database

### 3.1 Study area

The LP (640 000 km²) is a typical arid and semiarid area located in the middle reaches of the Yellow River (750 000 km²) and characterized by a continental monsoon climate. Elevations range from 84 m to 5207 m (Figure 2). The desert-steppe, typical steppe, and forest steppe (deciduous broad-leaf forest) zones are distributed from northwest to southeast, and correspond to mean annual isohyets of 250, 450, and 550 mm in the arid, semiarid, and semi-humid areas, respectively. The continuous loess covering ranges from 100 m to 300 m in thickness on the mountains, hills, basins, and alluvial plains of different heights. The northwestern part of the region is dominated by flat sandy areas. The middle and southeastern parts are characterized by EP induced water-erosion landform (Zhang et al., 1997), with a rugged undulating ground surface that is broken, barren, and dissected by gullies and ravines (Cai, 2001). EP-induced flooding episodes occur occasionally in the summer, with sediment concentrations generally exceeding 300 kg/m³ and have been observed to be as large as 1,240 kg/m³. The hyper-concentrated flooding has historically resulted in numerous disasters, with severe consequences to people and livestock (Zhang et al., 2017). The amount of soil erosion has been estimated to be larger than 2000 to 3000 million tons per year (Tang, 1990). Soil erosion has resulted in the density of gullies and ravines in the LP being larger than 3–4 km/km², with the maximum exceeding 10 km/km².

### 3.2 Database

To conduct the EP assessment, we used daily data available for 87 national meteorological stations in and around the LP (Figure 2b), consisting of continuous time series from 1961 to 2015. All the precipitation data were obtained from the China Meteorological Data Sharing Service System (http://cdc.nmic.cn/home.do). Missing data accounted for < 0.1% of total sample,





and were replaced by a value of 0 in this paper; this replacement of a very few missing values does not influence the analysis. Data regarding severity of soil erosion were provided by the LP Science Data Centre of the Data Sharing Infrastructure of Earth System Science of China (http://loess.geodata.cn). These data were compiled during the Soil Erosion Census of the First National Water Conservancy Census (2014). Mean annual vegetation coverage at 8 km spatial resolution and 15 day temporal resolution were computed using data for the period 1982 to 2006, produced by the Global Inventory Monitoring and Mapping

Studies (GIMMS) group from measurements of the advanced very high resolution radiometer (AVHRR) onboard the NOAA 7, NOAA 9, NOAA 11, and NOAA 14 satellites. Data from the National Center for Environmental Prediction/National Center for Atmospheric Research (NCEP/NCAR) Reanalysis Project (Kalnay et al., 1996) were also used in this study. The variables selected for analysis were the monthly mean geopotential height, monthly mean wind, daily mean sea level pressure and daily mean wind from 1961 to 2013 on a 2.5 °×2.5 ° spatial grid (http://www.esrl.noaa.gov/psd/).

**4 Results**

**4.1 Spatial characteristics of extreme precipitation**

Figure 3a shows that the mean annual precipitation (for the period 1961-2015) varied from 115 mm in the northwest to the 845 mm in the southeastern LP. The associated EPTs ranged from 17 mm/d in northwest to 50 mm/d from in southeast (Figure 3b); these EPT isohyets are generally consistent with these of mean annual precipitation. Figure 3b indicates that the area

around the Dongsheng station is a regional EP center since that the EPTs around the station are higher than these of the surrounding areas, whereas the isohyets of mean annual precipitation are smooth. The spatial distribution of MEPs is also similar to that of mean annual precipitation, increasing from northwest to southeast and ranging from 35 to 138 mm/yr, as shown in Figure 3c. Maximums of MEP occur in the southern and southeastern LP.

Figure 3d indicates that EPFs over the region during the past five decades have ranged from 54 days to 115 days (the annual

EPF ranged from 1.0 to 2.1). Notable occurrences of high EPF can be seen in and around the Ziwuling Mountains in the mid-southern LP, while the highest frequency occurred at the east of the Fenhe Valley in the southeastern LP. Meanwhile, the lowest frequency occurred in the northwestern LP, the western Muus sandy land, and the western Liupan Mountains.

Figure 3e indicates that the averaged EPIs ranged mainly between 0.3 and 0.7. The spatial variations of EPI and EPF contrast with each other, with highest EPIs centered in the mid-eastern LP, where EPFs were comparatively lower, and lowest EPIs in

the southeastern LP, which had the maximum mean annual precipitation and EPF. The highest values of EPI dominated the Central LP area. The northern boundary of the area was positioned southeast of the Muus Desert (Dongsheng and Yulin cities) and south of the Baiyu Mountains (Wuqi and Huanxian counties); the western boundary was positioned west of the Liupan Mountains (Haiyuan, Jingyuan and Yuzhong counties); the eastern boundary was positioned northeast of the Luliang Mountains and Fenhe Valley; the southern boundary was positioned to the north of the Central Shaanxi Plain. The EPI presents





the event-EP-power causing natural hazards. This high value of EPI in part explains why this area is characterized by very serious soil erosion that releases more than 2-billion ton sediment annually into channels of the Yellow River.

Figure 3f indicates that the spatial distribution of EPS in the LP increased from northwest to southeast, ranging from 0.27 to 0.66, but with the highest EPSs centered in the southeast Central LP. The areas with highest EPSs covered the basins of the Jing, Luo, and Fen rivers. Although an EP event always occurred over a small range, the spatial maps of EPI, EPF, and EPS

indicate that the areas with serious EP events are regularly distributed.

**4.3 Spatiotemporal variation of EP**

Our results (Figure 4a) indicate that 91.4% of the LP was characterized by a negative annual precipitation trend over the study period, whereas only 8.6% of the total area presented a positive trend. At the same time, the spatiotemporal trends of the annual EP ranged from -0.78 to +0.48 mm/yr (Figure 4b), with 23.8% of the total area showing a positive trend, with increased annual

EP distributed mainly in the southwestern LP (west of Lanzhou) and the mid-southern LP (Beiluo and Jing river basins and an area around the Xingxian station). Meanwhile, the annual EPF changed by -0.6 to +0.5 days, with a change rate ranging from $-1.2 \times 10^{-2}$ to $+0.95 \times 10^{-2}$ days/yr, as shown in Figure 4c. The areas with a negatively trending EPF covered 86.4% of the total area while the areas with positively trending EPF covered 13.6%, the latter occurring mainlyin the southwestern LP (around the Xining station) and in the areas around the Xi'an and Xingxian stations. The areas with notably decreasing trends

occurred mainly in the mid-west and southeast regions of the LP.

Figure 4d indicates that the changes of annual EPI ranged from -0.18 to +0.27, with a change rate ranging from $-0.34 \times 10^{-2}$ to $0.52 \times 10^{-2}$/yr. We found that 34 of the 87 stations showed a upward slope (6 stations with a significance level $p < 0.1$), and 53 stations (4 stations with a significance level $p < 0.1$) showed a negative slope. As shown in Figure 4d, areas with positive trends of EPI accounted for 42.2% of the total area, with the areas delineating by the Wulate-Yulin-Yan'an-Huashan

stations and the Jingtai-Xiji-Tianshui stations, as well as the area west of the Minhe station. The areas with a negative slope covered 57.8% of the total area.

Figure 4e indicates that the annual EPSs changed by -0.09 to +0.07 during the study period, with rates varying from $-0.34 \times 10^{-2}$ to $0.52 \times 10^{-2}$ $yr^{-1}$. Of the 87 stations, 39 stations showed a positive slope (3 stations with a significance level of $p < 0.05$), while 54 stations exhibited a negative slope (1 station with a significance level of $p < 0.05$). The areas with increased

EPSs covered 25.4% of the total area and were mainly found in an area delineated by the Wuqi, Tianshui, and Huashan stations and an area west of the Xiji station. The areas with negative trends accounted for 74.6% of the total area.

The trends estimates computed for annual EP, EPF, and EPI are associated with strong uncertainty. For instance, the upward trend of annual EP in and around the Xingxian station relied heavily on the upward trend of the EPF and not the downward trend of the EPI. The EPF around the Changwu station decreased, but both the annual EP and EPS increased with the upward

trend of the EPI. However, nearly all the areas with positive trends for annual EP, EPI, EPF, and EPS had a negative annual





precipitation (Figure 4). It should be noted that 62.1% of the LP area with negative annual precipitation has more than one positive EP index, potentially indicating the risk of more serious hazardous situations.

### 4.3 Intra-annual EP characteristics and their relationship to large-scale atmospheric circulation

### 4.3.1 The intra-annual Distribution of EP events

Figure 5 displays the intra-annual distributions of the MDP and the ADEP for the 87 stations from 1961 to 2015. Precipitation from June to September accounts for 72% of the total amount, while 91% of the total EP events occur from June to August. According to the fitted curve (Figure 5), the highest MDP occurred on July 26, which is 11 days earlier than the maximum ADEP on 6 August. Based on fitting the four-parameter Weibull curve ($p < 0.0001$), the MDP for the 224 days from March 26 to November 4 accounted for 95% of the mean annual precipitation. Meanwhile, the ADEP from May 21 to
September 18 accounted for 95% of the total EPF.

Therefore, high concentration of amount of daily precipitation into a limited period results in a significant alternation of wet and dry seasons in the LP. In addition, low precipitation but with annual alteration of dry and wet seasons, and highly concentrated intra-annual EP events with an occurrence 11 days earlier that the wettest days, contributes to a fragile eco-environment subject to severe natural hazards. Specifically, lower precipitation but highest EPI and EPS are responsible for
the most severe hazard situations in the Central LP, such as soil erosion.

### 4.3.2 Atmospheric circulation factors for the spatial variation of extreme precipitation

Atmospheric circulation is the leading factor causing the above phenomena. The LP is located in the East Asian monsoon region. According to the average sea level pressure and winds at the 1000 hpa level in winter from 1961 to 2015 (see Figure 6a), the dry winter in the region is influenced by the interactions between two high pressure areas in Southwest China (the
Tibet Plateau high pressure system) and North China (the Mongolia high pressure system). The prevailing East Asian winter monsoon (which has a north-northwest direction) circulates in East China and brings cold and dry airstreams. In contrast, the summer climate of the LP is affected by interactions between two high pressure systems, the Pacific high pressure and Tibet Plateau high pressure systems. Figure 6b shows that the prevailing East Asian summer monsoon (which has a south-southeast wind direction) brings warm and humid maritime airstreams that spread from the West Pacific to Central China. However, the
Tibet Plateau high pressure has a notable effect on the climate of the northwestern LP, and the airstream humidity decreases gradually as the distance from the Pacific increases. The resulting effect and decreased humidity is to form a vast arid region in Northwest China, including the northwestern LP, with a prevailing wind direction of west-southwest. This explains why precipitation decreases from southeast to northwest, and precipitation is scarce in the northwest LP.

Nevertheless, tropical cyclones occasionally enter the central LP, accompanied by EP events. For instance, in August 1996
a Western Pacific cyclone landed in the southeastern coastal area of China and weakened gradually as it moved northwest, as shown by the 1000-hPa geopotential height and winds in Figure 6c. Plenty of rainstorms or intense rainfall events accompanied





the cyclones occurred in its transit area. On 3 August 1996, the weakened cyclone reached the southeastern LP, as shown in Figure 6d. However, under the control of the Tibet Plateau high pressure, the central LP is generally the northwestern boundary to which the tropical cyclone can reach. As shown in Figure 6d, the cyclone was blocked from entering the northwestern LP,

moved towards the northeast, and gradually dissipated. These phenomena illustrates why this region has limited precipitation but severe EP events.

## 5. Discussion

### 5.1 Rationality of Spatial EP Characteristics

Natural hazards related to EP can be divided into two categories: (1) hazards accompanied by EP, and (2) hazards that follow

the occurrence of EP. For the former, one focus is the dependence of EP and storm surges in the coastal zone. Using such dependence structures, EP and storm surge can be quantified to provide information for successful hazard management (Svensson and Jones, 2004; Zheng et al., 2013). For the latter, the LP is such that the area suffers from EP-induced natural hazards that exceed the general tolerance of the natural environment, existing ecosystems, human life and social economy. In this case, the rational characteristics of EP responsible for spatial hazards can be studied.

Here, we use the widely distributed soil erosion to verify the rationality of our results. According to the universal soil loss equation (Wischmeier, 1976), the rational characteristics of EP should correlate well with soil erosion and vegetation coverage (Figure 7). To examine this, partial correlation analyses were performed between soil erosion intensity and EPI/EPS, and with the vegetation coverage. Our results indicate that water-based erosion intensity correlates significantly with vegetation coverage (negatively, Figure 7a) and EPS (positively, Figure 3f); the related coefficients are -0.61 ($p < 0.001$) and 0.53 ($p <$

0.001), respectively. For the correlations between water erosion intensity and vegetation coverage, and with EPI (Figure 3e), the coefficients are -0.58 ($p < 0.001$) and 0.76 ($p < 0.001$), respectively. This finding demonstrates the rationality of our results. Note that, the higher correlation between EPI and soil erosion agrees with the results of plot experiments by Tang (1993), who noted that high-intensity precipitation is the primary driving force of erosion.

Zhou and Wang (1992) divided the LP into three zones of raindrop kinetic energy (<1000, 1000–1500, and 1500–2000 J m$^{-2}$ yr$^{-1}$, respectively), based on observations of the raindrop kinetic energies of rainstorms during 1980s. We found that the 30

and 35 mm/d EPT contours closely overlap with the raindrop kinetic energy contours of 1000 and 1500 J m$^{-2}$ yr$^{-1}$. Further, soil erosion in the LP in recent decades has been found to be approximately 5000–10 000 t km$^{-2}$ yr$^{-1}$ (Ludwig and Probst, 1998; Shi and Shao, 2000). Such high rates of sediment erosion are generally induced by several rainstorm events during the year, with the top 5 daily sediment yields accounting for 70%–90% of the annual total soil loss (Rustomji et al., 2008; Zhang et al.,

2017). For instance, a 200-year precipitation event in Wuqi on 30 August 1994 induced a flooding event with a daily sediment concentration of 1060 kg/m$^3$. The streamflow was 2.41/25.6 times of the mean annual streamflow from 2002 to 2011, and the





sediment load was equivalent to 9.6% of the total sediment yields from 1963 to 2011 (Zhang et al., 2016). Therefore, it can be inferred that the EPF obtained in this study, about twice a year on average, is rational.

The highest EPF occurred 11 days earlier than the day of maximum daily precipitation in the LP (Figure 5). This means that, the days on which the LP experiences most serious EP events, tend to be days when precipitation is less. In other words, every year, the vegetation has not sufficiently recovered when the most frequent EP events occur in the LP. Further, the sparse spatial nature of precipitation is insufficient for the growth of high-coverage vegetation, especially in the northwestern area LP (Figure 7a). However, the highest EPI provides the strongest erosion force, which contributes to the severe rates of erosion (Figure 7b) in the Central LP. Given that 62.1% of the total LP with negative trend of annual precipitation has one or more positive EP indices, the underlying upward trends of water erosion and sediment yield should be taken into account in catchment management efforts.

## 5.2 Uncertainty in EP identification

The uncertainties in identification and assessment of EP events come from two aspects: (1) the stochasticity in climate (Miao et al., 2018) and (2) the methodology (Papalexiou et al., 2013). For the former, significant spatiotemporal variations occur in EP events as a result of varying geographical and meteorological conditions (Pinya et al., 2015). Extreme precipitation events are relatively rare, poorly predictable, and often with short duration, thus resulting in uncertainty in EP event identification. In the method section, the uncertainties in EPTs determination from paramedic and non-paramedic methods were discussed.

Figure 8 shows the results of EPF obtained by non-parametric methods for all 87 stations over the LP during 1961-2015. Large variances among the results, calculated at different percentile levels, are shown in Figures 8a–8c. It can be seen that the thresholds are smaller for lower percentiles. The standard deviation method provided similar results at different standard deviation levels (Figure 8f). As shown in Figures 8d–8e, a 50 mm/d threshold is probably suitable for the Southeast LP with higher mean annual precipitation, whereas a 25 mm/d threshold may be more suitable for some stations in the northwestern LP where there are no EP event exceeding 50 mm/d. Therefore, regardless of the varying geographical and meteorological conditions, the selection of these thresholds can be quite subjective and empirical. Note that the spatial causes for hazards situations cannot be theoretically explained by such methods.

Parametric methods require a predetermined threshold value, above which the data can be chosen as the EP series if the data series passed the goodness-of-fit test. As shown in Figure 9, both fixed values and percentiles were adopted to preset EPT. The selected rainfall series data were fitted to the gamma, GEV (generalized extreme value distribution), GPA (generalized Pareto distribution), Gumbel, exponential and the Weibull distributions, whose parameters were estimated with the $L$-moments method (Haddad et al., 2011) at a 0.05 significance level, using goodness-of-fit tests including K-S (Kolmogorov-Smirnov test), A-D (Anderson-Darling K-Sample test) and C-S (Pearson's Chi-squared test) tests. As shown in Figures 9a1-9a3 and 9b1-9b3, the results of the K-S and A-D tests are similar but different from those of C-S test for the preset fixed value and percentile thresholds.





Further, these results for different distribution functions are quite different from each other. As shown in Figures 9a1-9a2,
by the K-S and A-D tests, the passing rates from GEV and Gumbell distribution functions are high while there is almost no
passing rate from GPA, exponential and Gamma functions. In addition, the passing rates are different or evenly opposite
between preset methods of percentiles and fixed values. As shown in Figure 9, the GEV and Gumbell distribution function
have high passing rates for EP series obtained by preset percentiles (Figures 9a1-9a2), whereas the passing rate for these series
obtained by fixed values are very low (Figures 9b1-9b2). We also found that these distribution functions are not sensitive to
the percentile or fixed value changes. These findings indicates that the fitting accuracy can be greatly affected by the selection
of the extreme value distribution functions, goodness-of-fit tests and methods for EPT preset. Liu et al. (2013) noted that the
fitting accuracy is also affected by the size of rainfall series. So, unavoidably, applications of parametric methods also depend
on personal subjectivity and empiricism. These uncertainties may why prior studies of EP over the LP tend to disagree with
each other (Li et al., 2010b, 2012; Xin et al., 2009).

As noted by Pandey et al. (1998) and Douglas and Barros (2003), these methodological uncertainties arise due to the wide
gap between mathematical modeling and the physical understanding of precipitation processes. As previously mentioned, the
multifractal technique can be used to describe the statistical probability and physical processes associated with observed data
(Lovejoy and Schertzer, 2013; Tessier et al., 1996), while the scale invariance of multifractals enables the multifractal
technique to also overcome the influence of the sample size (Pandey et al., 1998; Tessier et al., 1996). Further, the segmentation
algorithm helps to overcome the problem of uncertainty. In the present study, the general correspondence and the specific
divergences between EPT and precipitation isohyets (Figure 3) further exhibits the varying meteorological and geographical
influences. Overall, the universal multifractal method provides a much superior approach to addressing uncertainties and
providing a unique set of EPTs.

## 6. Conclusions

We have proposed an approach that integrates universal multifractals with a segmentation algorithm to enable identification
of EP events, and thereby to assess its spatiotemporal EP characteristics in the LP, using data from 87 meteorological stations
from 1961 to 2015. We find that the spatial distribution of the EPTs increased from 17.3 mm/d in the northwestern to 50.3
mm/d in the southeastern LP. Similarly, the MEP increased from 35 mm to 138 mm/yr, with the maximum MEP occurred in
the southern and southeastern LP. The EPF over the LP was within a range of 54–116 days over the last 55 years. Notable
occurrences of EPFs mainly observed in the mid-southern and southeastern LP. An examination of atmosphere circulation
patterns demonstrates that the central LP is the boundary where tropical cyclones enter the inland China, resulting in the highest
EP intensity and EP severity in this area. Correlation analysis significantly supported the reasonability of the spatial estimates
of EP characteristics that are responsible for hazardous situations over the LP. The climate factors for the most serious
hazardous situations in the LP especially in the Central LP come from the low precipitation, the highest EPI and the highly
ADEP concentrated 11 days earlier than the wet season.



Spatiotemporally, annual EP increased in the southwestern and mid-southern LP. The areas with a positive EPF trend occurred in the southwestern LP and the areas around the Xi'an and Xingxian stations, whilst the areas with a positive trend of EPI among the Wulate-Yulin-Yan'an-Huashan stations and the Jingtai-Xiji-Tianshui stations, as well as the area west of the Minhe station. The annual EPSs with increased slope covered an area delineated by the Wuqi, Tianshui, and Huashan stations and an area west of the Xiji station. Overall, the areas with upward trends of the annual EP, EPF, EPI, and EPS accounted for 23.8%, 13.6%, 42.2%, and 25.4% of the LP area, respectively. It should be noted that 62.1% of the LP area with negative annual precipitation experienced upward trends of one or more EP variables. It can be concluded that EP over the LP intensified, potentially imposing a risk of more serious hazardous situation. Sustainable countermeasures should be considered in the catchment management to address the underlying hazards.

In conclusion, the universal multifractal approach considers both the physical processes and their probability distribution, and thereby provides an approach to overcome uncertainties and identify EP events without the need for empirical adjustments. This approach is thus useful for application to spatiotemporal EP assessment at regional scale.

*Data availability*. All the data used in this study are available upon request.

*Author contributions*. JZ, XZ and RL prepared the research project. JZ, HVG, GG and BF conceptualised the methodology. JZ developed the code and performed the analysis. JZ prepared the manuscript with contributions from all co-authors.

*Competing interests*. The authors declare that they have no conflict of interest.

**Acknowledgements:**

This research was funded by the National Natural Science Foundation of China (no. 41701207), the Open Foundation of the State Key Laboratory of Urban and Regional Ecology of China (SKLURE2019-2-5), and the Fundamental Research Funds for the Central Universities of China (2652018034, 2652018038). The second author acknowledges partial support by the Australian Centre of Excellence for Climate System Science (CE110001028). We thank the China Meteorological Data Sharing Service System, the Yellow River Conservancy Commission, the LP Science Data Centre of the Data Sharing Infrastructure of Earth System Science of China, and the NCEP/NCAR for providing data used in this paper. All data sources are publicly accessible and these websites are listed in Section 3.2.

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

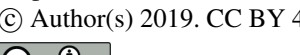



**Fig. 1: Procedure for the EPT determination of the Xingxian station. (a) The multifractal index α (black dots) and the alternative abrupt points (red dots). (b) Pooled variances (black dots) as calculated from the α series with the significant abrupt points (red dots) of the variances. (c) As in (a) but for the singularity $\gamma_S$. (d) As in (b) but for the variance calculated from the $\gamma_S$ series. (e)The time variation of daily precipitation ranging from 1961 to 2015. The blue dot line represents the determined EPT.**





Fig. 2: Location of the Loess Plateau in the middle reaches of the Yellow River, China (inset) and distribution of the meteorological stations in and around the LP.



Fig. 3: Spatial distributions of (a) EPTs, (b) MEP, (c) total EPF, (d) mean EPI, and (e) mean annual EPSI in the LP from 1961 to
2015.








**Fig. 4: The spatial distribution of the trends for (a) annual precipitation, (b) annual EP, (c) annual EPF, (d) annual EPI, and (e) annual EPSI in the LP from 1961 to 2015.**





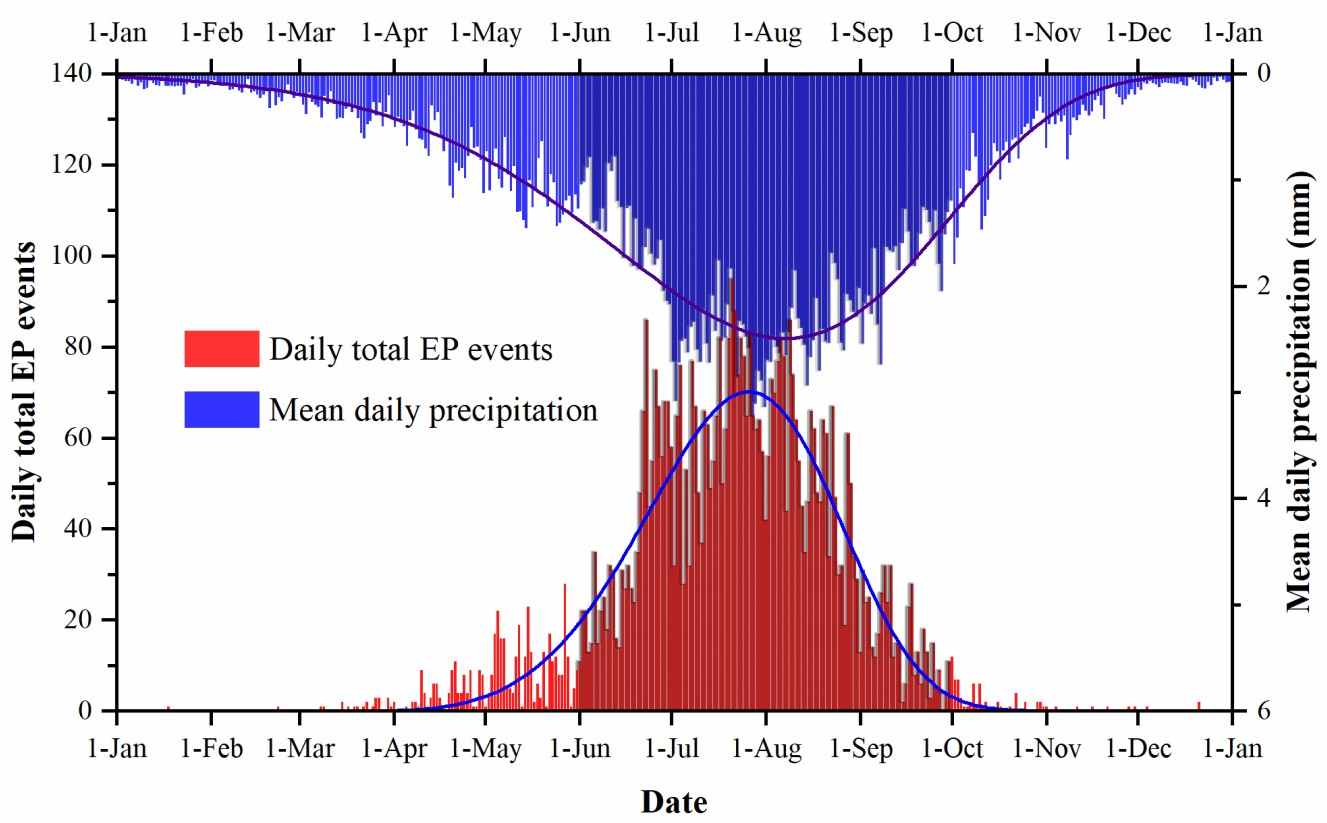

**Fig. 5: Intra-annual distribution of daily precipitation and the number of daily EP events for the 87 stations from 1961 to 2015 and**

**their fitting curves by Weibull function.**





**Figure 6: Average sea level pressure and winds: (a) the mean for all winters (from December to February) and (b) the mean for all summers (from June to August) from 1961 to 2015; Characteristics of 1000-hPa geopotential height and winds on (c) 1 August 1996 and (d) 3 August 1996. The data were derived from global NCEP/NCAR reanalysis average monthly and daily data.**






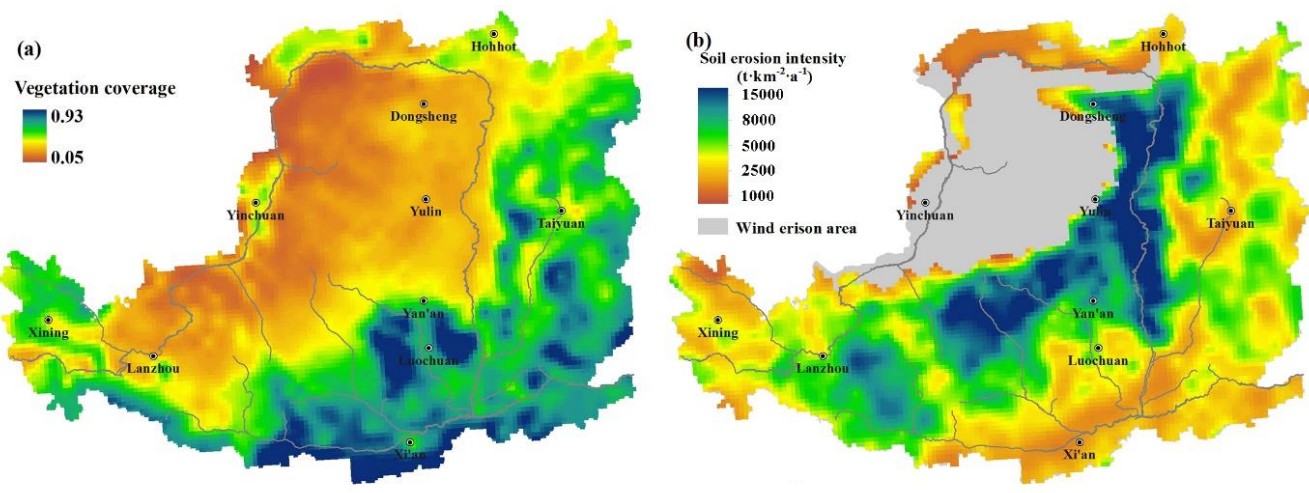

**Fig. 7: (a) Spatial distribution of mean vegetation coverage in summer (from June to August) on the Loess Plateau from 1982 to 2006 at a spatial resolution of 8 km. (b) Spatial distribution of the soil erosion intensity, which was resampled to a spatial resolution 8 km.**





**Fig. 8: EPTs determined by different methods and the corresponding EP frequencies for 87 stations over the Loess Plateau. The**
**abscissa represents the stations with an increase in mean annual precipitation from 104 mm to 918 mm. The signs in Figures 8a–8c**
**display EPT and EPF derived by the 95th, 99th and 99.5th percentiles, respectively, while the signs in Figures 8d–8f display the EPF**
**and the fixed thresholds 25 mm/d, 50 mm/d and the EPT derived by the three-time standard deviation method, respectively.**





**Figure 9: The passing rates of goodness-of-fit test for individual distribution functions, with EP data series selected by different preset thresholds. (a1) K-S test, (a2) A-D test and (a3) C-S test for different distribution functions using preset percentile thresholds. (b1) K-S test, (b2) A-D test and (b3) C-S test for different distribution functions using thresholds of fixed values. The significant level is 0.05. The symbol lines of the passing rate of Gumbell function in a1-a3 and those of the Exponential function in a1-a2 and b1-b3 were offset upward arbitrarily by 5 units, and the passing rate of Weibull function in a1-a2 were offset upward arbitrarily by 10 units to separate them.**




**Table 1: Indices (abbreviations) used in this study addressing precipitation variations.**

| Index | Definition | Units |
|-------|-----------|-------|
| EPT | extreme precipitation threshold | mm/d |
| MEP | mean annual extreme precipitation | mm |
| EPF | frequency of extreme precipitation event | d |
| EPI | intensity of extreme precipitation event | dimensionless |
| EPSI | severity of extreme precipitation event | dimensionless |
| MDP | long-term mean intra-annual daily precipitation | mm/d |
| ADEP | long-term accumulated intra-annual daily extreme precipitation events | d |