# Peer review of "A Universal Multifractal Approach to Assessment of Spatiotemporal Extreme Precipitation over the Loess Plateau of China"

_Hydrology and Earth System Sciences, 2019_

## Referee Comment (RC1) · Anonymous Referee #1 · 9 Oct 2019

Review_hess-2019-430

In the manuscript "A Universal Multifractal Approach to Assessment of Spatiotemporal Extreme Precipitation over the Loess Plateau of China". The authors try to proposal approach to identify the extreme precipitation events (EP), which is vital to the formation of soil erosion, and thereby to assess the spatiotemporal characteristics of the EP in the Loess Plateau (LP) during long term of 1961-2015. The study is interesting because providing new understanding about the spatiotemporal characteristics of EP in the LP. Meanwhile, the results are useful and contribute to the risk management (such as the soil erosion) in the LP. In my view, the MS may be suitable to be published in Hydrology

and Earth System Sciences after minor revision.

General comments: The paper is well written and the results are well presented. Bibliography very exhaustive. The analyzed dataset is interesting and the results can be useful to improve the knowledge of spatiotemporal characteristics of EP and could be potentially useful for risk management. The results show that the approach integrating the universal multifractal approach and segmentation algorithm based on parametric statistical method can be used to identify the EP events. Then, giving a detailed description about the spatiotemporal characteristics of EP. What's more, the rationality of EP results is explained and verified through the relationship between EP with the spatial characteristics of soil erosion. Therefore, the presented results are robust and add new knowledge on those relevant eco-hydrologic study and management.

Major comments: 1. Please explain the calculation of each indices of extreme precipitation in detail. It can help the readers to understand the meaning of the indices. Such as how to calculate the MEP?

2. L319-326: Please strengthen this paragraph. How do you get the condition about the recovery of vegetation from the EP intensity?

Specific comments 1. There is a lack of the specific meaning of EPT (threshold of extreme precipitation), please briefly explain it in the "Introduction".

2. Please include all the indices used in your study in Table 1. Such as EPS.

3. What is the difference of EPS and EPSI?

4. L 19: Please replace "scare" with "scarce".

5. L148: What is the specific criterion of "approximately equal" in the selection of EPT?

6. L159: Please explain the specific meaning of "n" in the equation.

7. L165: Is the PF represents EPF? Please illustrate the equation in detail.

8. L195: Why the missing data were replaced by a value of 0 in your paper?

9. L208: Please delete "from" after "50 mm/d".

10. L212: Please unify the unit of MEP between the manuscript (mm/y) and the Fig. 3 (mm/a).

11. L214-215: What do you mean "the annual EPF ranged from 1.0 to 2.1"?

12. L316-L306: What do you mean "The streamflow was 2.41/25.6 times of the mean annual streamflow from 2002 to 2011"? What is the meaning of "/" in your manuscript?

13. L317-318: Please explain "Therefore, it can be inferred that the EPF obtained in this study, about twice a year on average, is rational" in detail. What do you mean "twice a year"?

14. L336-338: Why 50 mm/d and 25 mm/d are the suitable threshold for the Southeast and Northwestern of LP, respectively?

15. L358: Please add "the reason" between "may" and "why"

16. Fig. 3: Please correct the title of the figure. There are 6 subfigures (a-f) in the figure 3. However, only 5 subfigures (a-e) are listed and explained in the title.

---

## Short Comment (SC1) · 19 Oct 2019

General comment

In this work the authors proposed an approach integrating the universal multifractals and a segmentation algorithm to precisely identify EP events. Then they assessed spatiotemporal variation of extreme precipitation over the Loess Plateau, China.

It is known that extreme precipitation in the Loess Plateau is one of the major agents causing serious environment hazards in the Loess Plateau. However, to my knowledge, traditional method including parameter methods and non-parameter method

could not gave such a rational result of extreme precipitation in both spatial and temporal distribution.

The method proposed in this paper is innovative, and the results are of great significant in catchment management. The paper is in good presentation and fluent expression, and the content of the paper suits to HESS.

My detailed comments follow:

1. The authors should describe the procedure to calculate the EP indices in detail.

2. Can the methodology proposed in this paper be applied to extreme precipitation assessment in many other regions? Please discuss this point at the end of the abstract.

3. Line 42: The most serious soil erosion in the Loess Plateau should be $3 \times 104 - 4 \times 104$ tǐǧkm-2Îǧyr-1. Please check it.

4. Line 153: The abbreviation of "EP severity (EPS)" doesn't agree with "EPSI" in Table 1. Please check it throughout the paper.

5. Line 215: Add the unit to the annual EPF.

6. Line 231: it should be "4.2 Spatiotemporal variation of EP".

7. Line 238: It should be "mainly in"

8. Please check the units used throughout the paper and make sure each unit is strictly unique in the paper. For example, Line 42 "yr-1", Line 214 "days" and Line 237 "days/yr" don't in consistent with "d" and "d/yr" used in Figures 3 and 4.

9. The scarce precipitation but intense EP is a major external agent inducing most serious sediment erosion in Loess Plateau. The spatial EPI and EPS derived in this paper is important as it well illustrate this question. This is an important question should be further discussed in section 5.1.

10. There are some studies to explore the EP in the Loess Plateau. The authors should

compare their results with previous studies.

---

## Referee Comment (RC2) · Anonymous Referee #2 · 10 Nov 2019

In this paper, the authors study observed extreme precipitation in the Loess Plateau in China, derived from 87 meteorological stations in the period 1961 to 2015. They find that while there was a decreasing trend in mean precipitation in general, the trend in extreme precipitation frequency, intensity, and severity was increasing in parts of the study area. They further find a correlation of extreme precipitation thresholds with soil erosion hazards that regularly happen in this area. They apply multifractal theory and a segmentation algorithm to derive thresholds of extreme precipitation, and state that this method is superior to non-parametric methods that use fixed absolute values or percentiles to define the extreme precipitation threshold. The structure of the paper and the language is clear.

[Figure]

This analysis in an area that is exposed to hazards related to extreme precipitation is certainly valuable. It further promotes an advanced statistical analysis method based on multifractal theory, that many potential readers are probably not familiar with, including myself. The presentation of the method is at some points confusing, and it has not become entirely clear to me what makes the multifractal method superior to the more common methods from reading the manuscript. I recommend that the authors could improve the manuscript in a major revision, by a better motivation and explanation of the analysis method, and by some additional analysis. In the following, I separate my comments into major and minor points.

Major points:

––––––––-

1. Many readers may be unfamiliar with the theory of multifractals, therefore I recommend to make the explanation in the Methods section somewhat more "didactial". I understand that it is only possible to provide a very brief outline of the theory in the paper, but I think it might be possible to present the method in a way that allows readers to grasp the general idea, and make them aware of this new method. The more interested reader can then be drawn to book by Lovejoy and Schertzer. Here are some specific issues:

1a) Eq. (1): What would be L and l in this specific case of station measurements? Why is lambda the density of stations? I thought you apply the method to each station individually, so I would rather expect it is something like the measurement interval?

1b) I don't really understand what "singularity" means in this context. Could you give a simple explanation in your own words, if this is possible in a few sentences? Which values can gamma take?

1c) Eq. (3): Is q an integer defining the order of the moment?

1d) Similar to 1b: Can you give some more explanation what the mulitfractal index

alpha means? For example, what does it mean if alpha<1 versus alpha>1? In Eq. (4) and Eq. (6) it looks like that alpha is written with a "'" ("prime"). Is this a typo?

1e) l. 135: Is the interval space d a parameter? How is it selected?

1f) Eq. (7): Define and explain mu_L, mu_R, s_D.

1g) Eq. (8): Should it be P(tau) instead of P(t) here? Otherwise I don't understand the meaning of this equation.

2. The trend calculation in section 4.3 is certainly an important result of your paper. Therefore, I strongly recommend to perform significant tests for ALL indices shown in Fig. 4, including the mean precipitation. Why did you use significance level p<0.1 for EPI, and p<0.05 for EPS? It would be better to use the same significance level for all indices. It would also be good to mark regions with significant trends in all panels. One possibility would be to mark all stations with positive significant trends with blue dots, all stations with negative significant trends with red dots, and all stations with insignificant trends with black dots.

3. I think the claim that the mutifractional method is superior to the more common analysis methods it is not yet clearly justified. There should be a direct comparison of the non-parametric methods with the multifractal method, especially for the results shown in section 5.2. In Fig. 8, can you add a panel with the EPTs calculated from the multifractal method, and explain the differences to the others? The "standard deviation method" shown in panel 8f comes out of nowhere, please define it. It is not explained anywhere yet. Could you also show the goodness-of-fit numbers for the EP distributions from the multifractal results, and compare them to the non-parametric methods shown in Fig. 9? You could mark them in the panels in Fig. 9, or list them in a table.

Minor points:

――――――-

4. Definition of the EP indices (Section 2.3): The abundance of symbols is confusing

here. The EP severity EPS is called EPSI in Table 1, Fig. 3f, and probably other places. Plase use the same acronyms everywhere. It is also confusing that EPI and EPT are called P_I and P_T, respectively, in Eq. (9). Better use EPI and EPT in Eq. (9). In Eq. (11), P_F is not defined. It is the same as EPF, I assume, so you can also better use EPF here.

5. Eq (11): How sensitive are the results for EPS to the choice of k1 and k2? (See Fig. 3f, Fig. 4e)

6. l. 84/85: "For precipitation, a scaling break ... roughly two weeks." What does this sentence mean? Could you be more precise?

7. The index EP as shown in Fig. 4b is not given in Table 1. Or do you mean MEP here?

8. l. 236: "... the annual EPF changed by -0.6 to +0.5 days,..." Where are these numbers from? They are not given in the figure. Is this the totlt trend for the whole time series?

9. l. 273: "According to the average sea level pressure and winds at the 1000 hpa level..." This does not seem to make sense. Either you can give the air pressure at a given height level, or the geopotential height at a given pressure level.

10. l. 334/335: "It can be seen... lower percentiles". This seems trivial. Either remove this sentence, or write something like: "Trivially, thresholds are smaller...."

11. Figs. 3 and 4: Why are there different station names shown in different panels? For example, the Xiji station is mentioned in connection with panel 4e, but it is not shown in this panel. So one has to find it in one of the other panels.

12. The tropical cyclone situation shown in Fig. 6c and 6d: Do these maps show mean fields for the whole day, or an instantaneous situation?

Small corrections:
* * *
l. 19: "scarce" (instead of "scare")

l. 167: "1" should be subscript in "k1".

l. 193: There is no Fig. 2b

l. 194: Link is not accessible to me.

l. 255: "... and EPS had a negative *trend in* annual..." and in the following line "... LP area with negative *trend in* annual..."

l. 332: "...parametric and non-parametric..."

---

## Referee Comment (RC3) · Anonymous Referee #3 · 23 Nov 2019

This paper suggests a novel approach to assess spatio-temporal extremes of precipitations and implements it over of the Loess Plateau of China. The topic is interesting and relevant for the community. The data used seems of quality. The framework of Universal Multifractals (UM) is appropriate for such an issue. However I would not recommend to publish this paper in its current state, mainly for methodological reasons.

Indeed the methodology developed to determine the EPT(section 2.2) seems to contradict the underlying ideas of a multifractal framework. If I understood well the suggested methodology, it consists in performing UM analysis on the series after removing more and more extremes (replacing them with which values ?). Then the retrieved parameters are analysed and a so called "physically meaningful" threshold determined.

I have trouble understanding the logic behind this choice. Indeed, the interest of UM analysis is to analyse the whole data available and obtain $K(q)$ and $c(gamma)$ which then fully characterize the variability across scales. Removing the extremes will simply degrade the quality of the scaling (hence the reliability of the estimates), bias the analysis, and not improve the knowledge on the studied series. EP should be derived directly from the co-dimension function or scaling moment function obtained on the best data available. gamma_s could actually be a good choice, but other could be developed notably to include notion of both intensity and frequency as suggested by the authors.

Since all the following depends on the the indicators obtained from this methodology, I believe that this methodology should either more justified (I may have miss a point) or updated before any further study.

In addition, indication of the quality of the scaling, and scaling curves should be provided to the reader.

---

## Author Comment (AC1) · 1 Dec 2019

**Reply to Anonymous Referee #3**

This paper suggests a novel approach to assess spatio-temporal extremes of precipitations and implements it over of the Loess Plateau of China. The topic is interesting and relevant for the community. The data used seems of quality. The framework of Universal Multifractals (UM) is appropriate for such an issue. However I would not recommend to publish this paper in its current state, mainly for methodological reasons.

Reply: Thank you for your attention reviewing our manuscript.

Indeed the methodology developed to determine the EPT(section 2.2) seems to contradict the underlying ideas of a multifractal framework. If I understood well the suggested methodology, it consists in performing UM analysis on the series after removing more and more extremes (replacing them with which values ?). Then the retrieved parameters are analysed and a so called "physically meaningful" threshold determined.

Reply: the eliminated "extremes" were replaced by zeros in the procedure of EPT determination.

I have trouble understanding the logic behind this choice. Indeed, the interest of UM analysis is to analyse the whole data available and obtain K(q) and c(gamma) which then fully characterize the variability across scales. Removing the extremes will simply degrade the quality of the scaling (hence the reliability of the estimates), bias the analysis, and not improve the knowledge on the studied series. EP should be derived directly from the co-dimension function or scaling moment function obtained on the best data available. gamma_s could actually be a good choice, but other could be developed notably to include notion of both intensity and frequency as suggested by the authors. Since all the following depends on the the indicators obtained from this methodology, I believe that this methodology should either more justified (I may have miss a point) or updated before any further study.

Reply: In universal multifractals analysis, $c(\gamma)$ is the statistical scaling exponent characterizing how its probability changes with scale (Lovejoy and Schertzer, 2013). In other words, the codimension function $c(\gamma)$ characterizes the sparseness of the $\gamma$-order singularities (maximum) (Tessier et al., 1994). The parameter $\gamma_s$, i.e. the singularity of a data set with finite sample size, represents the maximum of intensity ($\varphi_\lambda$) at the scale ratio $\lambda$ (Douglas and Barros, 2003). Therefore, $c(\gamma)$ and $\gamma_s$ naturally capture the statistical properties of extremes of a data set. Further, simulation with different $c(\gamma)$ or $\gamma_s$ give rise to great differences, especially in extremes (Lovejoy and Schertzer, 2007; Lovejoy and Schertzer, 2013).

Obviously, as it was commented, *Removing the extremes will simply change the scaling property of the data set*. If we gradually eliminate extreme precipitation (EP) (replacing extremes by zeros), these exponents or functions will change and

will sharply change if the majority of extremes are removed, because singularity largely depend on extremes (Lovejoy and Schertzer, 2007); and these abrupt point can be determined as extreme precipitation threshold (EPT). This is the theoretical basis for EPT determination, as shown in Figure 1 of the manuscript.

But the procedure determining EPT does not give rise to any bias in the following analysis, as the spatiotemporal variation of EP were analyzed using original data. Therefore, the methods in this study will not resulted in quality degradation etc., as it was concerned in this comment "*Removing the extremes will simply degrade the quality of the scaling (hence the reliability of the estimates), bias the analysis, and not improve the knowledge on the studied series*".

As it was commented "*EP should be derived directly from the co-dimension function or scaling moment function obtained on the best data available.*" It is sure that the multifractal representation captures the observations independently of when or how extreme precipitation came to be. In this way, the UM can be applied to infer the magnitude of precipitation maximum within a return period, and more precise results can be obtained in comparison with traditional parameter functions, see Douglas and Barros (2003). The first author had also applied the UM to estimate maximum precipitation with a duration in his doctoral dissertation, as shown in Figure 1 below. However, such an estimation of maximum precipitation within a duration has nothing to do with spatiotemporal EP variation assessment over a long period in a large area.

[Figure]

Figure 1. Projected extreme values as a function of their return period.

There are two techniques exploring multifractals: the UM and the multifractal detrended fluctuation analysis (MFDFA). The UM explores the characteristics of "extremes" while the MFDFA focuses on "normal variation". The MFDFA have been applied to determine EPT, see Du et al. (2013) and Liu et al. (2013). Motivated by these studies, we concluded that the UM, characterizing extremes, is appropriate to determine EPT. Thus, the method integrating UM and segmentation algorithm was proposed for EP assessment in the Loess Plateau, China. Besides, as the Short

Comments (SC1) noted, the results of spatial EP obtained in this study is much more rational at present.

Reference

Douglas, E.M., Barros, A.P., 2003. Probable maximum precipitation estimation using multifractals: application in the Eastern United States. Journal of Hydrometeorology, 4(6): 1012-1024.

Du, H., Wu, Z., Zong, S., Meng, X., Wang, L., 2013. Assessing the characteristics of extreme precipitation over Northeast China using the multifractal detrended fluctuation analysis. Journal of Geophysical Research: Atmospheres, 118: doi: 10.1002/jgrd.50487.

Liu, B., Chen, J., Chen, X., Lian, Y., Wu, L., 2013. Uncertainty in determining extreme precipitation thresholds. Journal of Hydrology, 503: 233-245.

Lovejoy, S., Schertzer, D., 2007. Scaling and multifractal fields in the solid earth and topography. Nonlinear Processes in Geophysics, 14(4): 465-502.

Lovejoy, S., Schertzer, D., 2013. The weather and Climate: emergent laws and multifractal cascades. Cambridge University Press.

Tessier, Y., Lovejoy, S., Schertzer, D., 1994. Multifractal analysis and simulation of the global meteorological network. Journal of applied meteorology, 33(12): 1572-1586.

In addition, indication of the quality of the scaling, and scaling curves should be provided to the reader.

Reply:According to this comment and comments from Anonymous Referee #2, the authors concluded that the part of methodology should be described in more detail.

---

## Author Comment (AC2) · 28 Dec 2019

**Reply to Anonymous Referee #1**

In the manuscript "A Universal Multifractal Approach to Assessment of Spatiotemporal Extreme Precipitation over the Loess Plateau of China". The authors try to proposal approach to identify the extreme precipitation events (EP), which is vital to the formation of soil erosion, and thereby to assess the spatiotemporal characteristics of the EP in the Loess Plateau (LP) during long term of 1961-2015. The study is interesting because providing new understanding about the spatiotemporal characteristics of EP in the LP. Meanwhile, the results are useful and contribute to the risk management (such as the soil erosion) in the LP. In my view, the MS may be suitable to be published in Hydrology and Earth System Sciences after minor revision.

Reply: Thank you very much for your favorite consideration and detailed suggestions. We have studied all the comments carefully and have made corrections. The responses below are the details of the plan to revise the manuscript.

**General comments**

The paper is well written and the results are well presented. Bibliography very exhaustive. The analyzed dataset is interesting and the results can be useful to improve the knowledge of spatiotemporal characteristics of EP and could be potentially useful for risk management. The results show that the approach integrating the universal multifractal approach and segmentation algorithm based on parametric statistical method can be used to identify the EP events. Then, giving a detailed description about the spatiotemporal characteristics of EP. What's more, the rationality of EP results is explained and verified through the relationship between EP with the spatial characteristics of soil erosion. Therefore, the presented results are robust and add new knowledge on those relevant eco-hydrologic study and management.

Reply: We quite appreciate your favorite consideration and insightful comments.

**Major comments:**

1. Please explain the calculation of each indices of extreme precipitation in detail. It can help the readers to understand the meaning of the indices. Such as how to calculate the MEP?

Reply: More detailed information about the calculation of each indices were introduced in the revision of the manuscript.

2. L319-326: Please strengthen this paragraph. How do you get the condition about the recovery of vegetation from the EP intensity?

Reply: Thanks for your suggestion. More detailed information and references were added in this section to cover this question.

**Specific comments**

1. There is a lack of the specific meaning of EPT (threshold of extreme precipitation), please briefly explain it in the "Introduction".

Reply: Thanks. We concluded that the specific meaning of EPT should be introduced in the revision of the manuscript.

2. Please include all the indices used in your study in Table 1. Such as EPS.

Reply: Sorry for my carelessness, EPS is EP severity.

3. What is the difference of EPS and EPSI?

Reply: Sorry for my carelessness, EPS is EP severity, and "EPSI" was deleted.

4. L 19: Please replace "scare" with "scarce".

Reply: Thanks for your reminding. The word was corrected.

5. L148: What is the specific criterion of "approximately equal" in the selection of EPT?

Reply: As shown in Figure 1, firstly, these abrupt points are the alternative EPT, and then the point and its corresponding variance with gentle slope on the right but deep slope on the left was determined as EPT in a station.

6. L159: Please explain the specific meaning of "n" in the equation.

Reply: The parameter $n$ was defined in the Revision.

7. L165: Is the PF represents EPF? Please illustrate the equation in detail

Reply: Yes. $P_F$ was used to present EPF in the manuscript. The parameter was replaced by its abbreviation, EPF, in the Revision.

8. L195: Why the missing data were replaced by a value of 0 in your paper?

Reply: Thanks for your detailed question. The very few missing data were replaced by zeros because precipitation days account for <20% in the Loess Plateau.

9. L208: Please delete "from" after "50 mm/d".

Reply: Thanks. It was corrected.

10. L212: Please unify the unit of MEP between the manuscript (mm/y) and the Fig. 3 (mm/a).

Reply: Thanks for your careful work. The unit of "yr" was uniformly used in the Revision to represent "year".

11. L214-215: What do you mean "the annual EPF ranged from 1.0 to 2.1"?

Reply: Sorry for my mistake. It should be "mean annual EPF". We corrected it in the Revision.

12. L316-L306: What do you mean "The streamflow was 2.41/25.6 times of the mean annual streamflow from 2002 to 2011"? What is the meaning of "/" in your manuscript?

Reply: Sorry for my careless. The characters "/25.6" should be deleted. I missed to deleted them before submitting.

13. L317-318: Please explain "Therefore, it can be inferred that the EPF obtained in this study, about twice a year on average, is rational" in detail. What do you mean "twice a year"?

Reply: We mean that, on average, there are about two EP events in each station. Maybe my expression is ambiguous. We rephrased this sentence.

14. L336-338: Why 50 mm/d and 25 mm/d are the suitable threshold for the Southeast and Northwestern of LP, respectively?

Reply: There is no precipitation event exceed 50 mm/d in the northwest Loess Plateau, where mean annual precipitation is <200 mm. However, precipitation events > 50 mm/d often occur in the southeast Loess Plateau. The EPTs determined by universal multifractals are less than 20 mm/d in some stations of the northwest Loess Plateau but > 50 mm/d in some stations of the southeast Loess Plateau. Therefore, our results demonstrated this viewpoint.

We have extended this paragraph to cover this comment.

15. L358: Please add "the reason" between "may" and "why"

Reply: The phrase was added.

16. Fig. 3: Please correct the title of the figure. There are 6 subfigures (a-f) in the figure 3. However, only 5 subfigures (a-e) are listed and explained in the title.

Reply: Thanks. Missed information was added.

---

## Author Comment (AC3) · 28 Dec 2019

**Reply to Anonymous Referee #2**

In this paper, the authors study observed extreme precipitation in the Loess Plateau in China, derived from 87 meteorological stations in the period 1961 to 2015. They find that while there was a decreasing trend in mean precipitation in general, the trend in extreme precipitation frequency, intensity, and severity was increasing in parts of the study area. They further find a correlation of extreme precipitation thresholds with soil erosion hazards that regularly happen in this area. They apply multifractal theory and a segmentation algorithm to derive thresholds of extreme precipitation, and state that this method is superior to non-parametric methods that use fixed absolute values or percentiles to define the extreme precipitation threshold. The structure of the paper and the language is clear.

Reply: Thank you very much for your favorite consideration and detailed suggestions. We have studied all the comments carefully and have made corrections. The responses below are the details of the plan to revise the manuscript.

This analysis in an area that is exposed to hazards related to extreme precipitation is certainly valuable. It further promotes an advanced statistical analysis method based on multifractal theory, that many potential readers are probably not familiar with, including myself. The presentation of the method is at some points confusing, and it has not become entirely clear to me what makes the multifractal method superior to the more common methods from reading the manuscript. I recommend that the authors could improve the manuscript in a major revision, by a better motivation and explanation of the analysis method, and by some additional analysis. In the following, I separate my comments into major and minor points.

**Major points:**

1. Many readers may be unfamiliar with the theory of multifractals, therefore I recommend to make the explanation in the Methods section somewhat more "didactial". I understand that it is only possible to provide a very brief outline of the theory in the paper, but I think it might be possible to present the method in a way that allows readers to grasp the general idea, and make them aware of this new method. The more interested reader can then be drawn to book by Lovejoy and Schertzer.

Reply: The methodology was introduced in more details.

**Here are some specific issues:**

1a) Eq. (1): What would be L and l in this specific case of station measurements? Why is lambda the density of stations? I thought you apply the method to each station individually, so I would rather expect it is something like the measurement interval?

Reply: Thanks for your detailed comment. The $\lambda$ is scale or resolution of the time series of observed precipitation. We mean that $\varphi_\lambda$ is precipitation intensity at scale $\lambda$, i.e. accumulated rainfall depth. I am sorry for my mistake in brackets in Line 103. $\lambda$ is indeed the measurement interval as you meant. Here, $l$ is the number of the embedded time series at scale $\lambda$. For example, for a data series of daily precipitation with length of $L = 1024$ days, if we defined $l = 128$, then

λ=8, $\varphi_\lambda$ is the maximum precipitation accumulated at 8 days.

The sentences were rewritten.

1b) I don't really understand what "singularity" means in this context. Could you give a simple explanation in your own words, if this is possible in a few sentences? Which values can gamma take?

Reply: The "singularity" means the maximum of precipitation in this paper, and generally, $\gamma_s$ >0. It will be explained in detail in the revision of this manuscript.

1c) Eq. (3): Is q an integer defining the order of the moment?

Reply: Yes, it was explained in the Revision.

1d) Similar to 1b: Can you give some more explanation what the mulitfractal index alpha means? For example, what does it mean if alpha<1 versus alpha>1? In Eq. (4) and Eq. (6) it looks like that alpha is written with a "'" ("prime"). Is this a typo?

Reply: The multifractal index, $\alpha$, quantifies the distance of the process from monofractality. When $\alpha = 0$, the process is monofractal, whereas $\alpha = 2$ means the divergence of data moments. For time series, $0 < \alpha < 1$. According to universal multifractal by Tessier et al. (1994) and Lovejoy and Schertzer (2013), $\alpha'$ is the multifractal index related to $\alpha$, $1/\alpha + 1/\alpha' = 1$.

These parameters were introduced in the Revision.

Lovejoy, S., Schertzer, D., 2013. The weather and Climate: emergent laws and multifractal cascades. Cambridge University Press.

Tessier, Y., Lovejoy, S., Schertzer, D., 1994. Multifractal analysis and simulation of the global meteorological network. Journal of applied meteorology, 33(12): 1572-1586.

1e) l. 135: Is the interval space d a parameter? How is it selected?

Reply: The parameter $d$ is an interval that was used to gradually remove extremes in the EPT determining procedure, and the $d$ was set 1 mm/d.

1f) Eq. (7): Define and explain mu_L, mu_R, s_D.

Reply: The parameters $\mu_L$, $\mu_R$, and $s_D$ were defined.

1g) Eq. (8): Should it be P(tau) instead of P(t) here? Otherwise I don't understand the meaning of this equation.

Reply: Yes, thanks for your careful work. The parameter was corrected as you noted.

2. The trend calculation in section 4.3 is certainly an important result of your paper. Therefore, I strongly recommend to perform significant tests for ALL indices shown in Fig. 4, including the mean precipitation. Why did you use significance level p<0.1 for EPI, and p<0.05 for EPS? It would be better to use the same significance level for all indices. It would also be good to mark regions with significant trends in all panels. One possibility would be to mark all stations with positive significant trends with blue dots, all stations with negative significant trends with red dots, and all stations with insignificant trends with black dots.

Reply: According to this comment, stations with significant trends were marked in Figure 4.

To show more stations with higher trends, the significance level of 0.1 was selected. The replotted Figure 4 is shown below

[Figure]

3. I think the claim that the mutifractional method is superior to the more common analysis methods it is not yet clearly justified. There should be a direct comparison of the non-parametric methods with the multifractal method, especially for the results shown in section 5.2. In Fig. 8, can you add a panel with the EPTs calculated from the multifractal method, and explain the differences to the others? The "standard deviation method" shown in panel 8f comes out of nowhere, please define it. It is not explained anywhere yet. Could you also show the goodness-of-fit numbers for the EP distributions from the multifractal results, and compare them to the non-parametric methods shown in Fig. 9? You could mark them in the panels in Fig. 9, or list them in a table.

Reply: A figure for EPT determined by multifractal method was added in Figure 8, as shown

below. The 3 times standard deviation method was briefly introduced. The goodness-of-fit of EP events determined by universal multifractals in individual stations show very high passing rates, 100%, as shown in the Table 1 below.

[Figure]

(a) 95th percentile          (e) 25 mm/d
(b) 99th percentile          (f) 50 mm/d
(c) 99.5th percentile        (g) Universal multifractals
(d) 3-time standard deviation

Tabel 1. Passing rates of goodness-of- fit test for EP events determined by universal multifractals method.

| Function | K-S test (%) | A-D test (%) | C-S test (%) |
|---|---|---|---|
| Weibull | 100 | 100 | 100 |
| GPA | 100 | 100 | 100 |
| GEV | 100 | 100 | 100 |
| Gumbell | 100 | 100 | 100 |
| Exponential | 100 | 100 | 100 |
| Gamma | 100 | 100 | 100 |

**Minor points:**

4. Definition of the EP indices (Section 2.3): The abundance of symbols is confusing here. The EP severity EPS is called EPSI in Table 1, Fig. 3f, and probably other places. Plase use the same acronyms everywhere. It is also confusing that EPI and EPT are called P_I and P_T, respectively, in Eq. (9). Better use EPI and EPT in Eq. (9). In Eq. (11), P_F is not defined. It is the same as EPF, I assume, so you can also better use EPF here.

Reply: Sorry for my carelessness. The acronym of EP severity was uniformly defined as EPS in the Revision. According to this comment, these parameters used in Eqs. (9) and (11) were replaced by their acronyms.

5. Eq (11): How sensitive are the results for EPS to the choice of k1 and k2? (See Fig. 3f, Fig. 4e)

Reply: The EPS is the combination of both EPF and EPI. Different values of k1 and k2 will result in different values of EPS in a pixel and hence the spatial variation of EPS. The values of k1 and k2 was set according to IPCC [2007].

6. l. 84/85: "For precipitation, a scaling break ... roughly two weeks." What does this sentence mean? Could you be more precise?

Reply: Studies of the scaling property of precipitation using multifractals showed that the scaling break of precipitation from one station does not always equal to those form the other stations around the world. Generally, the scaling break ranges from several days to about 1 month around the world, with an average about 2 weeks.

7. The index EP as shown in Fig. 4b is not given in Table 1. Or do you mean MEP here?

Reply: MEP is mean annual EP; it is used in spatial variation presentation. Figure 4b shows the trends of annual EP in spatial.

8. l. 236: "... the annual EPF changed by -0.6 to +0.5 days,..." Where are these numbers from? They are not given in the figure. Is this the totlt trend for the whole time series?

Reply: Yes, the "-0.6 to +0.5 days" were total changes of EPF over the past 55 years. These were calculated by multiplying slopes by years. The sentence was rewritten.

9. l. 273: "According to the average sea level pressure and winds at the 1000 hpa level..." This does not seem to make sense. Either you can give the air pressure at a given height level, or the geopotential height at a given pressure level.

Reply: Thanks for your suggestion. The sentence was rewritten.

10. l. 334/335: "It can be seen... lower percentiles". This seems trivial. Either remove this sentence, or write something like: "Trivially, thresholds are smaller...."

Reply: The sentence was rewritten as this comment suggested.

11. Figs. 3 and 4: Why are there different station names shown in different panels? For example, the Xiji station is mentioned in connection with panel 4e, but it is not shown in this panel. So one has to find it in one of the other panels.

Reply: These labels had been listed to describe the regions with different values or trends.

According to this comment, we have listed same station labels in all these figures.

12. The tropical cyclone situation shown in Fig. 6c and 6d: Do these maps show mean fields for the whole day, or an instantaneous situation?

Reply: Yes, these maps show mean fields for the whole day. This information was added.

**Small corrections:**

l. 19: "scarce" (instead of "scare")

Reply: The word was corrected.

l. 167: "1" should be subscript in "k1".

Reply: We corrected it.

l. 193: There is no Fig. 2b

Reply: The figure should be Fig. 2. We corrected it.

l. 194: Link is not accessible to me.

Reply: The website was changed to be http://data.cma.cn/. It was revised.

l. 255: "... and EPS had a negative *trend in* annual..." and in the following line "... LP area with negative *trend in* annual..."

Reply: Thanks. The sentences were corrected.

l. 332: "...parametric and non-parametric..."

Reply: Sorry for my carelessness. The two words were corrected.

---

## Author Comment (AC4) · 28 Dec 2019

**Reply to Referee Comment from Zhihua Shi**

**General comment**

In this work the authors proposed an approach integrating the universal multifractals and a segmentation algorithm to precisely identify EP events. Then they assessed spatiotemporal variation of extreme precipitation over the Loess Plateau, China. It is known that extreme precipitation in the Loess Plateau is one of the major agents causing serious environment hazards in the Loess Plateau. However, to my knowledge, traditional method including parameter methods and non-parameter method could not gave such a rational result of extreme precipitation in both spatial and temporal distribution. The method proposed in this paper is innovative, and the results are of great significant in catchment management. The paper is in good presentation and fluent expression, and the content of the paper suits to HESS.

Reply: Thanks a lot for your favorite consideration and detailed suggestions. We have studied all the comments carefully and have made corrections. The responses below are the details of the plan to revise the manuscript.

**My detailed comments follow:**

1. The authors should describe the procedure to calculate the EP indices in detail.

Reply: More detailed procedure to calculate EP indices was added.

2. Can the methodology proposed in this paper be applied to extreme precipitation assessment in many other regions? Please discuss this point at the end of the abstract.

Reply: Yes, the method can be applied to regional EP assessment in many other regions, it was introduced at the end of the abstract.

3. Line 42: The most serious soil erosion in the Loess Plateau should be $3\times10^4 - 4\times10^4$ t·km$^{-2}$·yr$^{-1}$. Please check it.

Reply: Thanks for your reminding. The data was corrected.

4. Line 153: The abbreviation of "EP severity (EPS)" doesn't agree with "EPSI" in Table 1. Please check it throughout the paper.

Reply: Thanks for your reminding. The EP severity was uniformly abbreviated as EPS in the Revision.

5. Line 215: Add the unit to the annual EPF.

Reply: Thanks for your reminding. The unit was added.

6. Line 231: it should be "4.2 Spatiotemporal variation of EP".

Reply: Thanks. The number of the section title was corrected.

7. Line 238: It should be "mainly in"

Reply: Thanks. The word was corrected.

8. Please check the units used throughout the paper and make sure each unit is strictly unique in the paper. For example, Line 42 "yr-1", Line 214 "days" and Line 237 "days/yr" don't in consistent with "d" and "d/yr" used in Figures 3 and 4.

Reply: We have checked all units used in the paper and made correction. The unit "d" and "yr" were adopted to represent day and year, respectively. Besides, units in the form of superscript negative powers instead of divided by positive powers were adopted although the paper.

9. The scarce precipitation but intense EP is a major external agent inducing most serious sediment erosion in Loess Plateau. The spatial EPI and EPS derived in this paper is important as it well illustrate this question. This is an important question should be further discussed in section 5.1.

Reply: Thanks for your suggestion. We have enlarged discussion in section 5.1 about EP events and their corresponding natural hazards including sediment erosion.

10. There are some studies to explore the EP in the Loess Plateau. The authors should compare their results with previous studies.

Reply: Comparison with previous studies was further discussed in section 5.1.

---

## Author Response (AR1)

**Subject: Revision of HESS-2019-430**

Dear Editors,

Thank you for your letter and for the reviewers' comments concerning our manuscript entitled "A Universal Multifractal Approach to Assessment of Spatiotemporal Extreme Precipitation over the Loess Plateau of China".

We quite appreciate your favorite consideration and the reviewer's insightful comments. These comments are all valuable and very helpful for revising and improving our paper, as well as the important guiding significance to our researches. We have studied all the comments from the tree anonymous reviewers and the short comments from Zhihua Shi carefully and have made corrections which we hope to meet with approval. The point-to-point responses to all the comments are as follows.

**Comments from Editor**

Please, address all referees' comment according to your reply, with special care for the method description which was the object of the more fundamental comments. Apart from the discussion about the method description and validity, the manuscript was appreciated and possibly it will be suitable for publication in HESS after revision.

Reply: Thank you very much for your favorite consideration. Both indices and methodology description were revised according to comments from all reviewers.

**Comments from Anonymous Referee #1**

In the manuscript "A Universal Multifractal Approach to Assessment of Spatiotemporal Extreme Precipitation over the Loess Plateau of China". The authors try to proposal approach to identify the extreme precipitation events (EP), which is vital to the formation of soil erosion, and thereby to assess the spatiotemporal characteristics of the EP in the Loess Plateau (LP) during long term of 1961-2015. The study is interesting because providing new understanding about the spatiotemporal characteristics of EP in the LP. Meanwhile, the results are useful and contribute to the risk management (such as the soil erosion) in the LP. In my view, the MS may be suitable to be published in Hydrology and Earth System Sciences after minor revision.

Reply: Thank you very much for your favorite consideration and detailed suggestions. We have studied all the comments carefully and have made corrections.

**General comments**

The paper is well written and the results are well presented. Bibliography very exhaustive. The analyzed dataset is interesting and the results can be useful to improve the knowledge of spatiotemporal characteristics of EP and could be potentially useful for risk management. The results show that the approach integrating the universal multifractal approach and segmentation algorithm based on parametric statistical method can be used to identify the EP events. Then, giving a detailed description about the spatiotemporal characteristics of EP. What's more, the rationality of EP results is explained and verified through the relationship between EP with the spatial characteristics of soil erosion. Therefore, the presented results are robust and add new knowledge on those relevant eco-hydrologic study and management.

Reply: We quite appreciate your favorite consideration and insightful comments.

**Major comments:**

1. Please explain the calculation of each indices of extreme precipitation in detail. It can help the readers to understand the meaning of the indices. Such as how to calculate the MEP?

Reply: More detailed information about the calculation of each indices were introduced in the revision of the manuscript.

See Lines 180-188, 201-204.

2. L319-326: Please strengthen this paragraph. How do you get the condition about the recovery of vegetation from the EP intensity?

Reply: Thanks for your suggestion. In this paragraph, we tried to discuss the intra-annual distribution of EP causing natural hazards in the LP. The day with highest EPF happens earlier than wettest day, indicating the climate reason why there is serious erosion in the LP. The paragraph was strengthened.

See Lines 360-367.

**Specific comments**

1. There is a lack of the specific meaning of EPT (threshold of extreme precipitation), please briefly explain it in the "Introduction".

Reply: Thanks. The specific meaning of EPT was introduced.

See Lines 65-67.

2. Please include all the indices used in your study in Table 1. Such as EPS.

Reply: Sorry for my carelessness. All indices were introduced.

See Lines 180-205.

3. What is the difference of EPS and EPSI?

Reply: Sorry for my carelessness, EPS is EP severity, and "EPSI" was deleted, as shown in Table 1.

4. L 19: Please replace "scare" with "scarce".

Reply: Thanks for your reminding. The word was corrected.

See Line 20.

5. L148: What is the specific criterion of "approximately equal" in the selection of EPT?

Reply: As shown in Figure 1, firstly, these abrupt points are the alternative EPT, and then the point and its corresponding variance with gentle slope on the right but deep slope on the left was determined as EPT in a station. The sentence was rephrased.

See Lines 175-177.

6. L159: Please explain the specific meaning of "n" in the equation.

Reply: The parameter *n* was defined.

See Line 183.

7. L165: Is the PF represents EPF? Please illustrate the equation in detail

Reply: Yes.  $P_F$  was used to present EPF in the manuscript. The parameter was replaced by its abbreviation, EPF, in the Revision.

See Lines 183-186.

8. L195: Why the missing data were replaced by a value of 0 in your paper?

Reply: Thanks for your detailed question. The very few missing data were replaced by zeros because precipitation days account for <20% in the Loess Plateau.

9. L208: Please delete "from" after "50 mm/d".

Reply: Thanks. It was corrected.

See Line 247.

10. L212: Please unify the unit of MEP between the manuscript (mm/y) and the Fig. 3 (mm/a).

Reply: Thanks for your careful work. The unit of "yr" was uniformly used in the Revision to represent "year".

See Line 251 and the others all through the paper.

11. L214-215: What do you mean "the annual EPF ranged from 1.0 to 2.1"?

Reply: Sorry for my mistake. It should be "mean annual EPF". We corrected it in the Revision.

Lines 253-254.

12. L316-L306: What do you mean "The streamflow was 2.41/25.6 times of the mean annual streamflow from 2002

to 2011"? What is the meaning of "/" in your manuscript?

Reply: Sorry for my careless. I missed to deleted the characters "/25.6" before submitting. The characters were deleted.

See Line 358.

13. L317-318: Please explain "Therefore, it can be inferred that the EPF obtained in this study, about twice a year on average, is rational" in detail. What do you mean "twice a year"?

Reply: It was corrected as "twice every year". As mentioned above, the top 5 daily sediment discharge account for 70%-90% of annual sediment discharge. The top 5 daily sediment discharge is generally produced by 1 or 2 EP events every year. The sentence was reworded.

Lines 359-360.

14. L336-338: Why 50 mm/d and 25 mm/d are the suitable threshold for the Southeast and Northwestern of LP, respectively?

Reply: There is no precipitation event exceed 50 mm/d in the northwest Loess Plateau, where mean annual precipitation is <200 mm. However, precipitation events >50 mm/d often occur in the southeast Loess Plateau. The EPTs determined by universal multifractals are less than 20 mm/d in some stations of the northwest Loess Plateau but > 50 mm/d in some stations of the southeast Loess Plateau. Therefore, our results demonstrated this viewpoint.

The sentences were rephrased.

See Lines 385-388.

15. L358: Please add "the reason" between "may" and "why"

Reply: The phrase was added.

See Line 414.

16. Fig. 3: Please correct the title of the figure. There are 6 subfigures (a-f) in the figure 3. However, only 5 subfigures (a-e) are listed and explained in the title.

Reply: Thanks. Missed information was added.

See Figure 3 in Line 612

**Comments from Anonymous Referee #2**

In this paper, the authors study observed extreme precipitation in the Loess Plateau in China, derived from 87 meteorological stations in the period 1961 to 2015. They find that while there was a decreasing trend in mean

precipitation in general, the trend in extreme precipitation frequency, intensity, and severity was increasing in parts of the study area. They further find a correlation of extreme precipitation thresholds with soil erosion hazards that regularly happen in this area. They apply multifractal theory and a segmentation algorithm to derive thresholds of extreme precipitation, and state that this method is superior to non-parametric methods that use fixed absolute values or percentiles to define the extreme precipitation threshold. The structure of the paper and the language is clear.

**Reply: Thank you very much for your favorite consideration and detailed suggestions. We have studied all the comments carefully and have made corrections.**

This analysis in an area that is exposed to hazards related to extreme precipitation is certainly valuable. It further promotes an advanced statistical analysis method based on multifractal theory, that many potential readers are probably not familiar with, including myself. The presentation of the method is at some points confusing, and it has not become entirely clear to me what makes the multifractal method superior to the more common methods from reading the manuscript. I recommend that the authors could improve the manuscript in a major revision, by a better motivation and explanation of the analysis method, and by some additional analysis. In the following, I separate my comments into major and minor points.

Reply: Thanks for your wonderful work.

**Major points:**

1. Many readers may be unfamiliar with the theory of multifractals, therefore I recommend to make the explanation in the Methods section somewhat more "didactial". I understand that it is only possible to provide a very brief outline of the theory in the paper, but I think it might be possible to present the method in a way that allows readers to grasp the general idea, and make them aware of this new method. The more interested reader can then be drawn to book by Lovejoy and Schertzer.

Reply: Thanks for your reminding. The methodology was introduced in more details.

See Lines 106-148.

**Here are some specific issues:**

1a) Eq. (1): What would be L and l in this specific case of station measurements? Why is lambda the density of stations? I thought you apply the method to each station individually, so I would rather expect it is something like the measurement interval?

Reply: Thanks for your detailed comment. I am sorry for my mistake. The  $\lambda$  is the scale ratio of the time series of observed precipitation, and it is indeed the measurement interval as you meant.  $\varphi_{\lambda}$  is accumulated precipitation at scale  $\lambda$ . Here, l is the number of the embedded time series at scale  $\lambda$ . For example, for a data series of daily precipitation with length of L = 1024 days, if we defined the measurement scale l = 16 days, then  $\lambda=64$ ,  $\varphi_{\lambda}$  is the maximum precipitation accumulated at 16 days.

See Lines 106-108.

1b) I don't really understand what "singularity" means in this context. Could you give a simple explanation in your own words, if this is possible in a few sentences? Which values can gamma take?

Reply: The "singularity" means the maximum of precipitation at scale ratio  $\lambda$  in this paper, and generally,  $\gamma_s > 0$ . It was explained in detail in the revision of this manuscript.

See Lines 108, 132-133.

1c) Eq. (3): Is q an integer defining the order of the moment?

Reply: Yes, it was explained in the Revision.

See Lines 112-113.

1d) Similar to 1b: Can you give some more explanation what the mulitfractal index alpha means? For example, what does it mean if alpha<1 versus alpha>1? In Eq. (4) and Eq. (6) it looks like that alpha is written with a "'" ("prime"). Is this a typo?

Reply: The multifractal index,  $\alpha$ , quantifies the distance of the process from monofractality. When  $\alpha = 0$ , the process is monofractal, whereas  $\alpha = 2$  means the divergence of data moments. For time series,  $0

Figure 1. Projected extreme values as a function of their return period.

- Douglas, E.M., Barros, A.P., 2003. Probable maximum precipitation estimation using multifractals: application in the Eastern United States. Journal of Hydrometeorology, 4(6): 1012-1024.
- Lovejoy, S., Schertzer, D., 2013. The weather and Climate: emergent laws and multifractal cascades. Cambridge University Press.
- Tessier, Y., Lovejoy, S., Schertzer, D., 1994. Multifractal analysis and simulation of the global meteorological network. Journal of applied meteorology, 33(12): 1572-1586.

In addition, indication of the quality of the scaling, and scaling curves should be provided to the reader.

Reply: One of the aims in this paper is to propose a method to determine EPT using multifractal technique. The scaling properties of precipitation, runoff etc., were widely explored using multifractal methods in the past 15 years. There are many papers can be referred. For clarification, the methodology was introduced in more detail.

See Lines 106-178.

**Short Comments from Zhihua Shi**

**General comment**

In this work the authors proposed an approach integrating the universal multifractals and a segmentation algorithm

to precisely identify EP events. Then they assessed spatiotemporal variation of extreme precipitation over the Loess Plateau, China. It is known that extreme precipitation in the Loess Plateau is one of the major agents causing serious environment hazards in the Loess Plateau. However, to my knowledge, traditional method including parameter methods and non-parameter method could not gave such a rational result of extreme precipitation in both spatial and temporal distribution. The method proposed in this paper is innovative, and the results are of great significant in catchment management. The paper is in good presentation and fluent expression, and the content of the paper suits to HESS.

Reply: Thanks a lot for your favorite consideration and detailed suggestions. We have studied all the comments carefully and have made corrections.

**My detailed comments follow:**

1. The authors should describe the procedure to calculate the EP indices in detail.

Reply: Procedure to calculate individual EP indices was added.

See Lines 180-205.

2. Can the methodology proposed in this paper be applied to extreme precipitation assessment in many other regions? Please discuss this point at the end of the abstract.

Reply: Yes, the method can be applied to regional EP assessment in many other regions, it was introduced at the end of the abstract.

See Lines 28-29.

3. Line 42: The most serious soil erosion in the Loess Plateau should be  $3 \times 10^4 - 4 \times 10^4 \text{ t} \cdot \text{km}^{-2} \cdot \text{yr}^{-1}$ . Please check it.

Reply: Thanks for your reminding. The data was corrected.

See Line 43.

4. Line 153: The abbreviation of "EP severity (EPS)" doesn't agree with "EPSI" in Table 1. Please check it throughout the paper.

Reply: Thanks for your reminding. The EP severity was uniformly abbreviated as EPS in the Revision.

See Line 189, Table 1 and the others all through the paper.

5. Line 215: Add the unit to the annual EPF.

Reply: Thanks for your reminding. The unit was added.

See Line 254.

6. Line 231: it should be "4.2 Spatiotemporal variation of EP".

Reply: Thanks. The number of the section title was corrected.

See Line 272.

7. Line 238: It should be "mainly in"

Reply: Thanks. The word was corrected.

See Line 281.

8. Please check the units used throughout the paper and make sure each unit is strictly unique in the paper. For example, Line 42 "yr-1", Line 214 "days" and Line 237 "days/yr" don't in consistent with "d" and "d/yr" used in Figures 3 and 4.

Reply: We have checked all units used in the paper and made correction. The unit "d" and "yr" were adopted to represent day and year, respectively. Figures 3 and 4 was replotted and the units were corrected. Besides, units in the form of superscript negative powers instead of divided by positive powers were adopted all through the paper.

See Figure 3 3 and 4.

9. The scarce precipitation but intense EP is a major external agent inducing most serious sediment erosion in Loess Plateau. The spatial EPI and EPS derived in this paper is important as it well illustrate this question. This is an important question should be further discussed in section 5.1.

Reply: Thanks for your suggestion. We have enlarged discussion in section 5.1 about EP events and their corresponding natural hazards including sediment erosion.

See Lines 362-372.

10. There are some studies to explore the EP in the Loess Plateau. The authors should compare their results with previous studies.

Reply: Further discussed was added in 5.2.

See Lines 411-415.

**A Universal Multifractal Approach to Assessment of Spatiotemporal Extreme Precipitation over the Loess Plateau of China**

Jianjun Zhang1, 2, 5, Guangyao Gao2, Bojie Fu2, Cong Wang2, Hoshin V. Gupta3, Xiaoping Zhang4 and Rui Li4

[revised manuscript text omitted]